

# Reconstructing the archosaur radiation using a Middle Triassic archosauriform tooth assemblage from Tanzania

Devin K. Hoffman[1], Hunter R. Edwards[1,2], Paul M. Barrett[3] and Sterling J. Nesbitt[1]

[1] Department of Geosciences, Virginia Polytechnic Institute and State University (Virginia Tech), Blacksburg, VA, USA
[2] Department of Geological Sciences, University of Cape Town, Rondebosch, South Africa
[3] Department of Earth Sciences, Natural History Museum, London, UK

Corresponding author
Devin K. Hoffman, devinkh5@vt.edu

## ABSTRACT

Following the Permo–Triassic mass extinction, Archosauriformes—the clade that includes crocodylians, birds, and their extinct relatives outside crown Archosauria—rapidly diversified into many distinct lineages, became distributed globally, and, by the Late Triassic, filled a wide array of resource zones. Current scenarios of archosauriform evolution are ambiguous with respect to whether their taxonomic diversification in the Early–Middle Triassic coincided with the initial evolution of dietary specializations that were present by the Late Triassic or if their ecological disparity arose sometime after lineage diversification. Late Triassic archosauriform dietary specialization is recorded by morphological divergence from the plesiomorphic archosauriform tooth condition (laterally-compressed crowns with serrated carinae and a generally triangular lateral profile). Unfortunately, the roots of this diversification are poorly documented, with few known Early–Middle Triassic tooth assemblages, limiting characterizations of morphological diversity during this critical, early period in archosaur evolution. Recent fieldwork (2007–2017) in the Middle Triassic Manda Beds of the Ruhuhu Basin, Tanzania, recovered a tooth assemblage that provides a window into this poorly sampled interval. To investigate the taxonomic composition of that collection, we built a dataset of continuous quantitative and discrete morphological characters based on in situ teeth of known taxonomic status (e.g., *Nundasuchus*, *Parringtonia*: $N = 65$) and a sample of isolated teeth ($N = 31$). Using crown heights from known taxa to predict tooth base ratio (= base length/width), we created a quantitative morphospace for the tooth assemblage. The majority of isolated, unassigned teeth fall within a region of morphospace shared by several taxa from the Manda Beds (e.g., *Nundasuchus*, *Parringtonia*); two isolated teeth fall exclusively within a "*Pallisteria*" morphospace. A non-metric multidimensional scaling ordination ($N = 67$) of 11 binary characters reduced overlap between species. The majority of the isolated teeth from the Manda assemblage fall within the *Nundasuchus* morphospace. This indicates these teeth are plesiomorphic for archosauriforms as *Nundasuchus* exhibits the predicted plesiomorphic condition of archosauriform teeth. Our model shows that the conservative tooth morphologies of archosauriforms can be differentiated and assigned to species and/or genus, rendering the model useful for identifying isolated teeth. The large overlap in tooth shape among the species present and their overall

similarity indicates that dietary specialization lagged behind species diversification in archosauriforms from the Manda Beds, a pattern predicted by Simpson's "adaptive zones" model. Although applied to a single geographic region, our methods offer a promising means to reconstruct ecological radiations and are readily transferable across a broad range of vertebrate taxa throughout Earth history.

## INTRODUCTION

Adaptive radiations, or evolutionary diversifications, play a critical role in the history of life as clades speciate and fill new ecological roles over geologically rapid time intervals (*Simpson, 1944*; *Schluter, 1996*). Although there are examples of adaptive radiations that are not speciose (e.g., Darwin's finches), or adaptively disparate (e.g., crotaphytine and oplurine iguanids), such a framework is still useful for structuring macroevolutionary questions and explaining present (and past) biological diversity (*Gavrilets & Losos, 2009*). Adaptive radiations and the shifts in evolutionary rates associated with them are among the most studied aspects of evolutionary biology (*Stanley, 1979*; *Losos & Miles, 1994*, *2002*; *Gavrilets & Losos, 2009*; *Revell et al., 2018*; *Slater & Friscia, 2019*). However, empirical uncertainties remain regarding many of the properties of adaptive radiations (*Gavrilets & Losos, 2009*; *Slater & Friscia, 2019*), with the relative timings of lineage diversification and ecological disparity during adaptive radiations being one such problem. Does lineage diversification come first, followed by specialization and evolution within an "adaptive zone" (*Simpson, 1944*, *1953*) or does ecological specialization drive lineage diversification simultaneously (*Schluter, 1996*)? In the former case species fill the same resource zones using similar, ancestral morphological structures (e.g., identical tooth morphologies), whereas in the latter each species would be expected to have a unique, derived morphology for its resource zone at the start of the radiation (for an empirical example, see *Slater & Friscia, 2019*). Determining which of these competing hypotheses operated in a case requires us to reconstruct an evolutionary radiation where a species-poor, adaptively restricted clade diversifies into a species-rich, adaptively disparate clade.

One such radiation occurred in the Triassic Period, following the Permo–Triassic mass extinction (PTME; *Raup, 1979*; *Erwin, 1994*; *Chen & Benton, 2012*; *Benton & Newell, 2014*) as archosauriforms recovered, rapidly diversified, and spread across Pangea to dominate terrestrial ecosystems for the next 150 million years (*Nesbitt, 2011*; *Ezcurra & Butler, 2018*). In addition to Archosauriformes being a speciose and disparate radiation, they also provide an opportunity to test adaptive radiations at a higher phylogenetic level. Lineage diversification of archosauriforms was rapid after the PTME, and by the Middle Triassic many archosauriform clades had appeared (*Ezcurra, 2016*; *Foth et al., 2016*; *Ezcurra & Butler, 2018*). By the Late Triassic, archosauriforms, including the crown group Archosauria, filled a wide variety of ecological roles, from top predators to large herbivores, and were represented in terrestrial, freshwater, and even marine ecosystems

(*Li et al., 2006*; *Butler et al., 2019*). If lineage diversification occurs first, followed by subsequent ecological disparity, we would expect Middle Triassic archosauriforms from across the tree to represent a limited range of ecologies. The question then arises, how can we best measure ecological disparity? Ecological disparity covers a variety of physiological, behavioral, and morphological traits, but the nature of the fossil record limits its measurement primarily to morphology. Previous work has used cranial morphology as a measure of disparity (*Foth et al., 2016*); however, complete, or even partial, skulls are rare for Early–Middle Triassic archosauriforms. Therefore, an alternative morphological system to approximate ecological disparity is needed. In this study, we use teeth as an indicator of ecological disparity because they have relatively high preservation potential (*Turner-Walker, 2008*) and offer a direct link to ecology through diet (*Lucas, 1979*; *Dessem, 1985*; *Scanlon & Shine, 1988*; *Sander, 1997*; *Linde, Palmer & Gómez-Zurita, 2004*; *Santana, Strait & Dumont, 2011*; *Zahradnicek et al., 2014*; *Melstrom & Irmis, 2019*). Quantitative analyses of tooth morphology have previously been used to document fossil assemblages with an emphasis on diet (*Larson, 2008*; *Frey & Monninger, 2010*; *Larson & Currie, 2013*; *Hendrickx, Mateus & Araújo, 2015*; *Larson, Brown & Evans, 2016*; *D'Amore et al., 2019*; *Melstrom & Irmis, 2019*). We consider diet as the aspect of ecology of interest in this study because the relative ease of its inference from morphology alone and the use of diet in previous studies of evolutionary radiations (*Slater & Friscia, 2019*).

Although tooth assemblages are rare in Middle Triassic terrestrial strata, recent fieldwork (2007, 2008, 2012, 2015, 2017) in the Manda Beds of the Ruhuhu Basin, Tanzania (*Sidor & Nesbitt, 2017*), has revealed a rich assemblage of archosauriforms known from postcrania and partial crania, including teeth (*Nesbitt et al., 2010*, *2014*; *Smith et al., 2018*). Specifically, these teeth come from the middle and upper Lifua Member bone accumulations (*Smith et al., 2018*), except one tooth (NMT RB831) which comes from the lower bone accumulation (*Nesbitt et al., 2017*; *Smith et al., 2018*), which are thought to be Anisian in age (*Rubidge, 2005*) but may be as young as early Carnian (*Ottone et al., 2014*; *Marsicano et al., 2016*; *Peecook et al., 2018*; *Wynd et al., 2018*). If the Anisian age is correct, then this is one of the oldest diverse archosauriform faunas known that is also represented by specimens from historical collections (*Butler et al., 2010*, *2018*; *Nesbitt et al., 2010*, *2013*, *2014*, *2017*; *Barrett, Nesbitt & Peecook, 2015*). Because this assemblage preserves members of ecological and phylogenetically diverse archosauriform lineages (e.g., Dinosauriformes and Loricata) and is chronologically between Early Triassic lineage diversification and Late Triassic ecological diversification it represents an excellent candidate for testing the relative timing of ecological and lineage diversification in the archosauriform radiation. Using a combination of information from these new and historical collections, we quantify tooth disparity in this earliest part of the archosauriform radiation to generate a morphospace visualization. From this we can assign isolated teeth to specific taxon, visualize inter- and intraspecific variation as well as intra-individual variation, and use this variation as a window into the ecological disparity of the archosauriforms within the Lifua Member assemblage. To achieve these goals, we use a combination of in situ teeth from jaw elements assignable to particular species

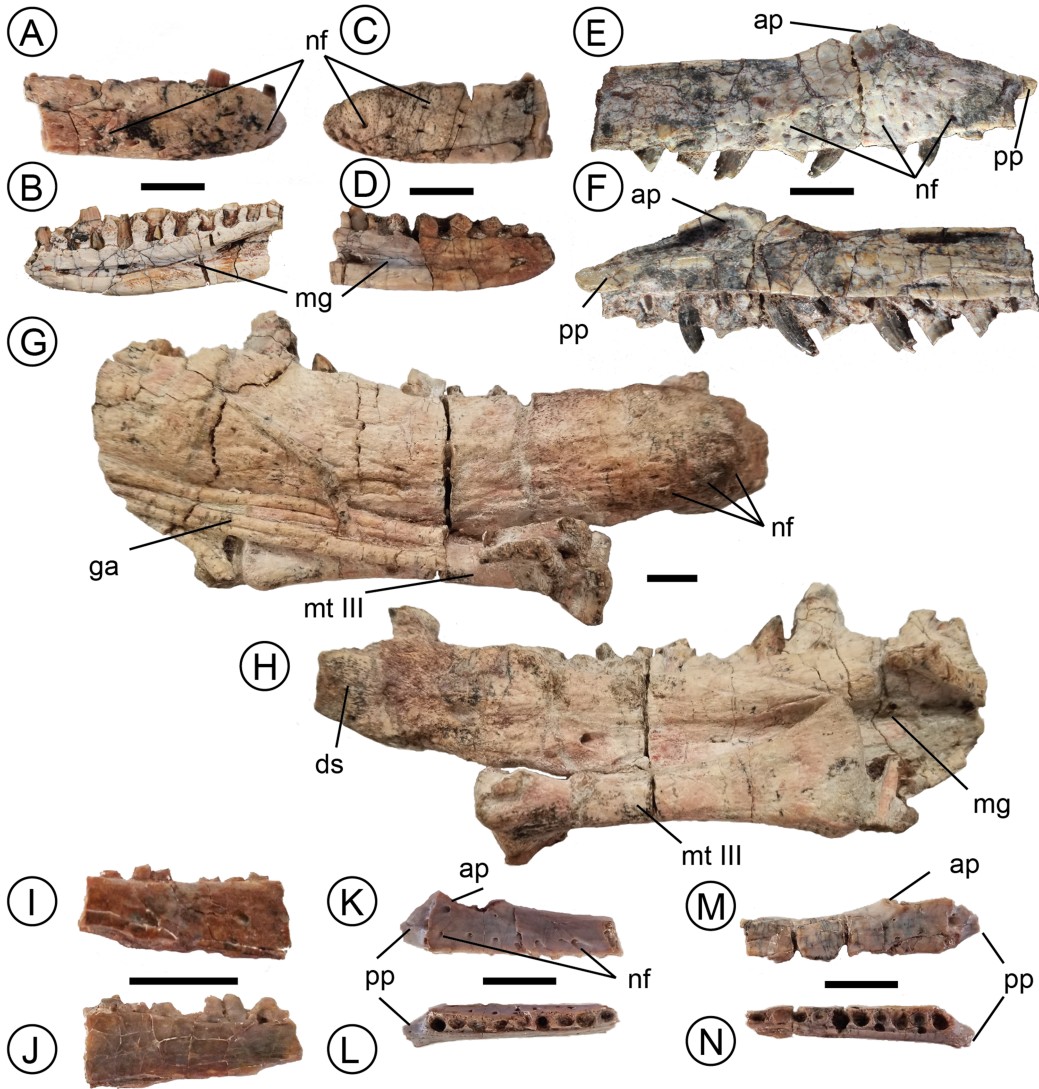

**Figure 1 A sample of the in situ dental material used for baseline measurements in this study.** *Parringtonia* (NMT RB426) left dentary in (A) lateral and (B) medial (bottom) views and (C) right dentary in lateral and (D) medial views. Undescribed archosauriform taxon (NMT RB187) right maxilla in (E) lateral and (F) medial view. *Nundasuchus* (NMT RB48) holotype right dentary in (G) lateral and (H) medial views. *Asilisaurus* (NMT RB 159) right dentary in (I) lateral and (J) medial views, (K) left maxilla in lateral and (L) occlusal views, and (M) right maxilla in lateral and (N) occlusal views. Abbreviations: ap, ascending process of the maxilla; ds, dentary symphysis; ga, gastralia; mg, Meckelian groove; mt III, metatarsal III; nf, nutrient foramen; pp, palatal process. All scale bars one cm.

(Fig. 1), and isolated teeth attributable to Archosauriformes (Fig. 2). Of particular interest is whether the isolated teeth fall within or expand the region of morphospace, and therefore, feeding ecologies occupied by the described Manda Beds taxa.

## MATERIALS AND METHODS

The 31 isolated teeth included in this study were collected from surface accumulations during fieldwork in 2007, 2008, 2012, 2015, and 2017 from the Manda Beds of the Ruhuhu

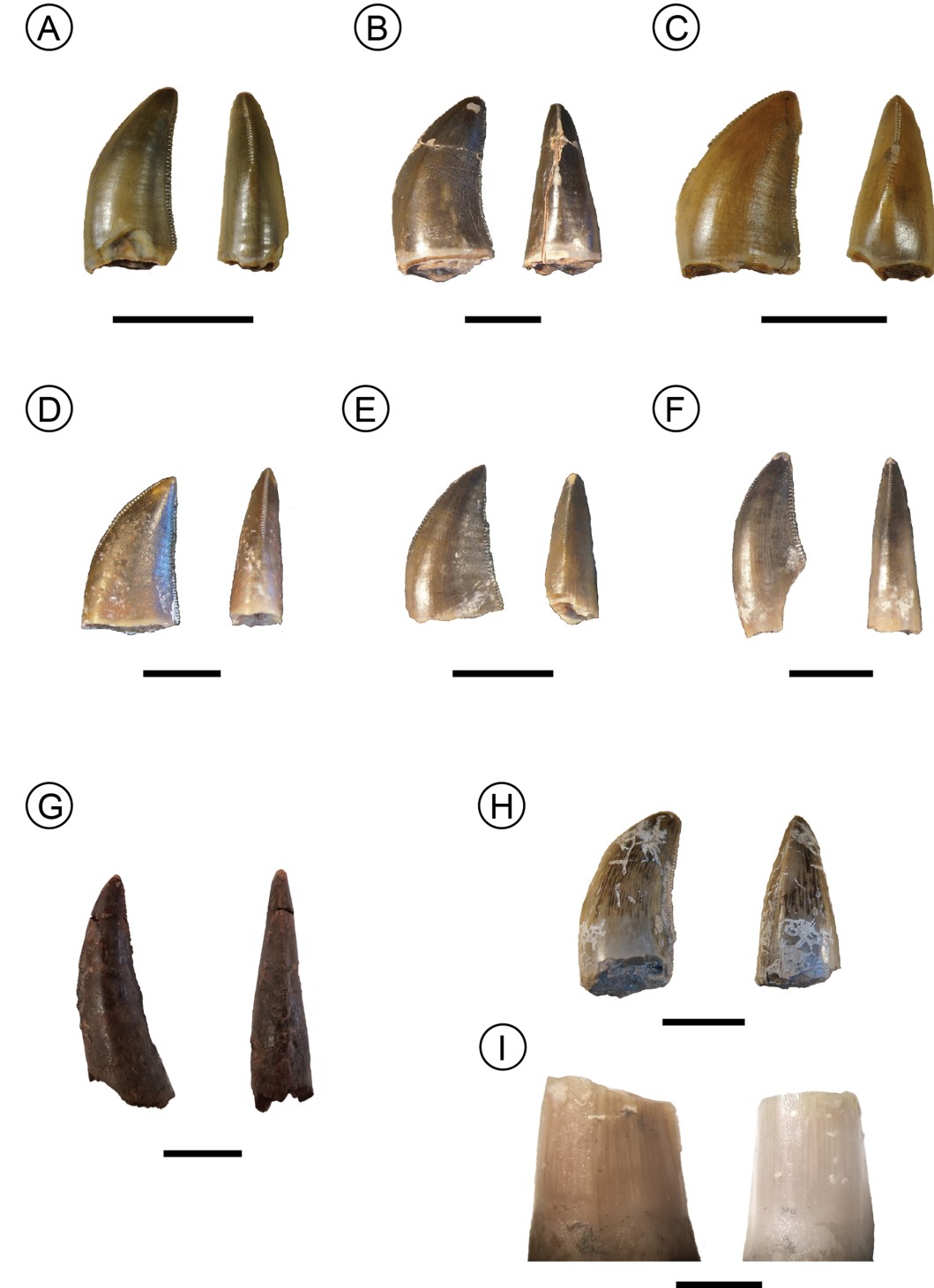

**Figure 2 Examples of isolated teeth from the Manda Beds tooth assemblage.** Morphotype A specimens from left to right (A) NMT RB807, (B) NMT RB827, (C) NMT RB809 in lateral and mesial views. Morphotype B, specimens from left to right, (D) NMT RB810, (E) NMT RB819, (F) NMT RB811 in lateral and mesial views. (G) Sole representative of Morphotype C NMT RB831 in lateral and mesial views. (H and I) Isolated teeth of known taxa. (H) *Nundasuchus* NMT RB48. (I) *Parringtonia* NMT RB426 in lateral and mesial views. Scale bars (A–C) one cm; (D–F) five mm; (G) one cm; (H) one cm; (I) two mm.

Basin by a multi-institutional team (*Sidor & Nesbitt, 2017*). All of the isolated teeth included in this study are currently housed at Virginia Tech Department of Geosciences and will be permanently reposited and managed in the National Museum of Tanzania. In addition to these isolated teeth (seven of which were referred to *Nundasuchus*, see *Nesbitt et al., 2014*), we also included teeth from within the tooth-bearing elements of five taxonomically distinct archosauriforms from the Manda Beds: *Nundasuchus* (NMT RB48), *Parringtonia* (NMT RB426), *Asilisaurus* (NMT RB159), "*Pallisteria*" (NHMUK PV R36620), and one currently undescribed pseudosuchian that we refer to by its specimen number (NMT RB187). We assign the isolated teeth to Archosauriformes based on their serration morphology (*Nesbitt, 2011*, character 128, states 1 & 2) as well as their general ziphodont construction, including lateral compression (*Godefroit & Cuny, 1997*). The Manda Beds tooth assemblage also includes teeth of cynodonts, temnospondyls, and other reptiles. Because we limited this study to teeth that we can confidently identify to Archosauriformes, we may have excluded unusual archosauriform teeth that do not resemble their "typical" condition. Therefore, our reconstruction of disparity should be considered a minimum estimate.

To quantify tooth shape, linear measurements (total crown height, base width, and fore-aft base length) and denticle counts were made following the protocol in *Smith, Vann & Dodson (2005)*, although due to the smaller size of the teeth in our study, we used one mm denticle densities, rather than five mm densities (Data S1). All statistical analyses were performed in R (v 3.1.2) and the RStudio console (v 1.1.383). All graphs of quantitative data were made using the R package "ggplot2" (*Wickham, 2009*). To capture tooth disparity (from log-transformed linear measurements), we used sum of variances with 95% predictive intervals following the methodology of *Larson, Brown & Evans (2016)*. We chose to use sum of variances as our measure of disparity owing to its prevalence in the literature and its robustness when working with small sample sizes. Sample size varied from 2 to 14 teeth; however, sample size does not significantly affect the sum of variance analysis (*Ciampaglio, Kemp & McShea, 2001*). The 14 teeth of *Asilisaurus* largely contributed base and width measurements because only three teeth were complete enough to measure crown height. We constructed a linear model in R that predicts the variable of tooth base shape (ratio of mesiodistal length over labiolingual width) by the variables of total crown height and species-level assignment (= base shape = total crown height × species assignment). The effects of each species on predicting tooth base shape were elucidated using the R package "lsmeans" (*Lenth, 2016*) using a pairwise comparison in the model by taxon. We plotted the teeth of known taxonomic affinity using ggplot2 (*Wickham, 2009*) to produce a base morphospace into which we plotted results from the isolated teeth for comparison.

Simple quantitative measurements only capture the general form of the teeth, and all of the teeth in this study resemble the hypothetical ancestral archosauriform tooth (serrated, recurved, and laterally compressed: *Nesbitt, 2011*). In order to more fully capture and describe the subtle variation of these teeth, a method of capturing discrete variation is needed. Non-metric multidimensional scaling (NMDS) is an ordination method that visualizes variation that can incorporate discrete qualitative features. We created a set of

**Table 1 Discrete character descriptions.** Summary of the discrete, binary traits used for scoring teeth in the NMDS analysis.

| | Description |
|---|---|
| 1 | Tooth apex, location, relative to the distal margin of the tooth base: tip mesial to or in the same vertical plane as the distal edge (0) or tip is located more distal than the distal edge (= recurved) (1). |
| 2 | Tooth lingual/labial, surfaces: texture is smooth (lack of crenulations, ridges, etc.) (0) or surface texture possess a series of parallel ridges from tooth apex to base (= fluted) (1). |
| 3 | Tooth labial/lingual, shape: crown curvature unequal (one side expanded relative to other) (0) or equal labial and lingual curvature (1). |
| 4 | Mesial tooth margin, shape: curvature angles change gradually (0) or angle changes abruptly at a single discrete point along mesial edge (1). |
| 5 | Tooth crown, size: labiolingual widths dorsal to the tooth crown base are all less than the crown base width (0) or a crown labiolingual width dorsal to the tooth crown base is greater than the crown base width (1). |
| 6 | Mesial/distal crown margins, surfaces: denticle caudae (= grooves on crown surface from between individual denticles) are absent (0) or present (1) (from *Abler, 1992*). |
| 7 | Mesial margin, length: mesial denticle row ends at a point sub-equal with distal denticle row (0) or mesial denticle row ends significantly further apically on crown than distal row (1). Can only be scored for teeth with both mesial and distal denticle series. |
| 8 | Mesial/distal margins, denticle density: number of mesial and distal denticles is <3 per mm (0), or ≥3 per mm (1). Measurements are taken near the middle of the carina. |
| 9 | Mesial margin, location: vertical axis of the mesial carina is in line the mesial-distal long axis (0) or laterally offset from the mesial distal long axis (1). |
| 10 | Mesial/distal margins, size: average size of mesial and distal denticles are the same (0) or the average size of the mesial and distal denticles is different (1). |
| 11 | Mesial/distal margins, shape: lateral profile shape of mesial and distal denticles remains constant (0) or denticles' lateral profile changes shape (e.g., rounded to square) (1). |

11 binary characters for scoring isolated and in situ teeth for NMDS (Table 1; Fig. 3; Data S2). All characters except one are new to this analysis (trait 6 "dental caudae = shallow grooves extending from between two adjacent denticles present/absent" is taken from *Abler (1992)*). In order to avoid circular reasoning when comparing our ecological signal to taxonomic and clade identity, we selected traits that have not been used in phylogenetic analyses of archosauriforms previously in the same form. The NMDS analysis was conducted in palaeontological statistics (PAST) (*Hammer, Harper & Ryan, 2001*) with a Bray–Curtis transformation. We ran an additional NMDS analysis in PAST using average taxon and morphotype scores where traits were scored for each taxon with >50% agreement of in situ teeth. Traits for which <50% of the specimens in the taxon or morphotype had the same score were scored as unknown ("?").

# ISOLATED TOOTH DESCRIPTIONS

## Morphotype A

These teeth (Figs. 2A–2C) are generally triangular in outline in lateral view and most are recurved (the point of the crown is distal to the distal-most extent of the crown base) although the remainder have crown tips that are level with the distal-most extent of the crown base. The labial and lingual sides of the crown lack ridges (i.e., no fluting), and the labial side of the crown exhibits greater convexity than the lingual side. The mesial denticle series terminates more apically along the crown margin than the distal series, which continues along the entire height of the crown though both start at the tip of the crown.

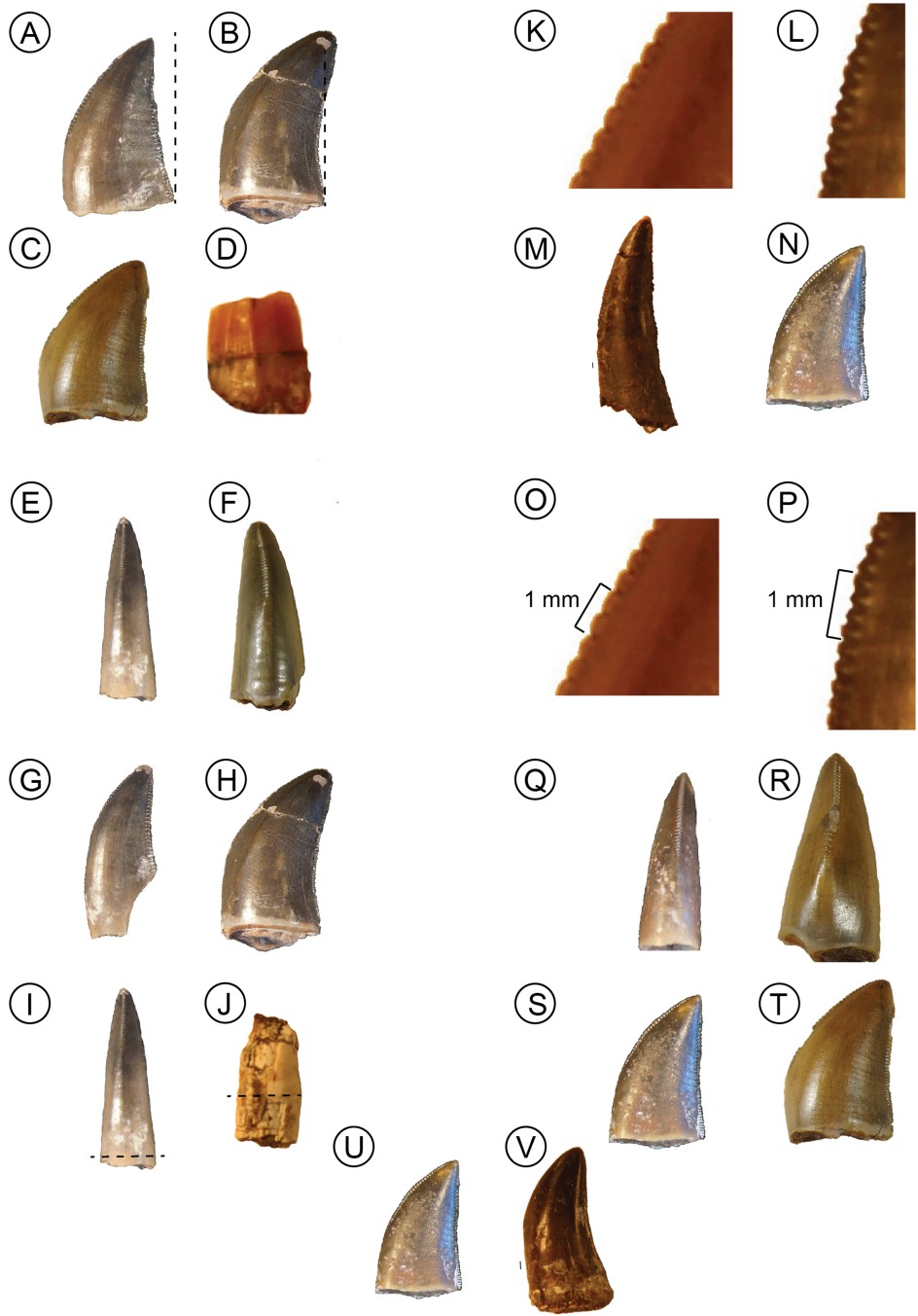

**Figure 3 Visualization of discrete traits.** In all traits score 0 on left and score 1 on right. Trait 1, degree of recurvature, (A) NMT RB819 and (B) NMT RB827. Trait 2, fluting, (C) NMT RB809 and (D) NMT RB426. Trait 3, labiolingual curvature, (E) NMT RB811 and (F) NMT RB819. Trait 4, mesial margin angle, (G) NMT RB811 and (H) NMT RB827. Trait 5, labiolingual bulge, (I) NMT RB811 and (J) NMT RB48. Trait 6, dental caudae, (K) NMT RB809 and (L) NMT RB810. Trait 7, mesial vs distal serration series length, (M) NMT RB831 and (N) NMT RB810. Trait 8, denticle density per mm, (O) NMT RB809 and (P) NMT RB810, black lines equal one mm. Trait 9, mesial margin alignment, (Q) NMT RB810 and (R) NMT RB809. Trait 10, mesial vs distal denticle density, (S) NMT RB810 and (T) NMT RB809. Trait 11, denticle shape variation along crown, (U) NMT RB810 and (V) NMT RB48.

The mesial denticle series is also offset from the mesial-distal long axis of the crown base, deflecting to the lingual side toward the crown base. The denticle densities range from two to five per mm. Denticle caudae (*Abler, 1992*), which are shallow grooves extending from between two adjacent denticles, are often present and directed parallel to the denticles. These denticle caudae are most easily observed in mesial or distal view (Fig. 3L).

In general, Morphotype A teeth strongly resemble both in situ and isolated teeth of *Nundasuchus* (Fig. 1; *Nesbitt et al., 2014*). Particularly important features are the presence of denticle caudae, an unequal labial-lingual curvature, and the more apical termination of the mesial denticle series relative to the distal denticle series. Also like *Nundasuchus*, Morphotype A teeth exhibit a mix of states in the changing curvature of the mesial crown edge in lateral view, with some teeth gradually changing angles and others exhibiting an abrupt shift in angle. The in situ teeth of *Nundasuchus* can exhibit either state depending on the proximity of the mesial edge of the crown to the distal edge of the preceding tooth. Although this combination of traits is only found in *Nundasuchus* in the Manda Beds fauna, archosauriforms from the Middle and Late Triassic elsewhere possess the same traits (*De Oliveira & Pinheiro, 2017*; *Schoch et al., 2018*).

## Morphotype B

These teeth (Figs. 2D–2F) are triangular in lateral view and are occasionally recurved, although in most the apical tip of the crown is approximately level with the distal-most end of the crown base. Morphotype B tooth crowns lack fluting and, in contrast to Morphotype A, the labial and lingual curvatures are equal. None of the teeth are bulbous (no labiolingual measurements are greater than crown base width). In the majority of Morphotype B teeth the mesial margin of the crown possesses a single point where the angle of the mesial carina changes abruptly. As in Morphotype A teeth, the mesial series of denticles in Morphotype B teeth terminates on the crown further apically than the distal series, which often terminates at the crown base. However, the mesial row of denticles is in line with the mesial-distal long axis of the crown base. The denticle densities range from three to eight per mm. Denticle caudae are present on some of the teeth and are directed parallel to the denticles. Although these teeth bear a strong resemblance to Morphotype A, they can be differentiated by their equal labial and lingual curvatures. Morphotype B teeth are similar to some of the in situ and isolated *Nundasuchus* teeth (Figs. 1 and 2; *Nesbitt et al., 2014*).

## Morphotype C

This morphotype (Fig. 2G) is represented by a single tooth in our assemblage, NMT RB831. The overall shape is tall, near conical, and recurved. The crown lacks fluting and the labial curvature is greater than the lingual curvature. Although its labial-lingual curvature is unequal, the mesial denticle series is positioned along the midline of the mesial-distal long axis. The orientation of the mesial edge of the tooth changes gradually, forming a long, continuous curve. The tooth is not bulbous. Denticle densities range from two to four per mm, and no denticle caudae are present. There is no variation in either the shape or size of the denticles between the mesial and distal series or along the length of

the crown. Unlike either Morphotypes A or B, the mesial series of denticles in Morphotype C ends at approximately the same level on the crown as the distal series, just above the crown base.

In general size and shape, as well as in many of its discrete features, the Morphotype C tooth is similar to the teeth of "*Pallisteria*" based on our observations. The teeth of the latter taxon are large, conical, recurved, and possess unequal labial-lingual curvature. The denticle density is low (<3 per mm) in the middle part of the tooth crown and denticles show little variation in shape or size. Unfortunately, none of the "*Pallisteria*" teeth could be scored for Trait 7 (termination height of the mesial denticle series; Table 1) due to poor preservation of the denticles, which otherwise differentiates the Morphotype C tooth from morphotypes A and B. If Morphotype C is like, or is, "*Pallisteria*," then subsequent "*Pallisteria*" tooth discoveries should be expected to have sub-equally extending mesial and distal denticle rows.

## IN SITU TOOTH DESCRIPTIONS

### *Nundasuchus*

We included a total of 13 *Nundasuchus* teeth (six in situ and seven isolated) originally described in *Nesbitt et al. (2014)*. The teeth range in height from 5.6 to 22 mm with denticle densities from two to five per mm. All of the teeth are labio-lingually compressed and are serrated on both mesial and distal margins. Only two teeth (NMT RB48A, NMT RB48E) possess a recurved tip that extends past the distal-most end on the tooth base (Fig. 3). Most teeth are smooth on the sides with a single exception exhibiting fluting (NMT RB48E: Fig. 3). All of the teeth possess unequal labial-lingual curvatures, a mesial row of denticles that terminates higher on the tooth crown than the distal row of denticles, and a mesial carina that is offset from the midline. Only two of the teeth possess dental caudae (NMT RB48A, NMT RB48G) and one tooth is bulbous (NMT RB48C: Fig. 3). In some teeth the mesial and distal denticle rows differ in size and/or in shape. About half the teeth have a distinct point on the mesial margin where the angle of the edge changes abruptly. For the in situ teeth, this seems to be related to how close the tooth is to the preceding socket, with the closer the distance being associated with an abrupt angle shift point.

### *Asilisaurus*

We included 14 in situ teeth though only three of these included more than the very base of the tooth. These three ranged in height from 1.6 to 2.9 mm and had a denticle density of approximately eight per mm. The teeth are closely packed, ankylosed to the sockets, and peg-like in shape (*Nesbitt et al., 2010*). All of the teeth have: smooth sides, equal labial-lingual curvature, and subeven mesial and distal row of denticles. None of the *Asilisaurus* teeth possess dental caudae and the mesial edge of the teeth changes angles gradually.

### *Parringtonia*

Of the 14 teeth in the study, 12 were in situ and the other two larger, isolated teeth. The teeth range in size from 2.5 to 21.6 mm, though the tallest in situ tooth is 8.3 mm, and

**Table 2 Results of linear model (base ~ total crown height + taxon).** All measures of significance are calculated in reference to the intercept, *Asilisaurus*. Therefore, while the undescribed pseudosuchian and *Parringtonia* can be differentiated in the model from *Asilisaurus*, the interrelationships are unknown.

|  | Estimate | Standard error | *t*-value | *p*-value |
|---|---|---|---|---|
| *Asilisaurus* (intercept) | 1.1285 | 0.1011 | 11.156 | <0.0001 |
| Total crown height (mm) | 0.0059 | 0.0035 | 1.714 | 0.0933 |
| Undescribed | 0.3718 | 0.1148 | 3.237 | 0.0022 |
| *Nundasuchus* | 0.0995 | 0.1247 | 0.798 | 0.4290 |
| "*Pallisteria*" | −0.1249 | 0.2007 | −0.622 | 0.5369 |
| *Parringtonia* | 0.2490 | 0.1127 | 2.210 | 0.0321 |

the denticle densities vary from 5 to 15 per mm. Most of the *Parringtonia* teeth lacked crown tips, though the two complete teeth are not recurved (Fig. 2I). All of the teeth are labio-lingually compressed and possess fluting and a mesial carina along the midline. The mesial and distal denticle series of all the teeth remain constant in both shape and size, though the mesial denticle series terminates higher on the crown than the distal series. In all the teeth the mesial edge angle changes gradually.

### NMT RB187

All 13 teeth of the teeth included from NMT RB187 are in situ. The labio-lingually compressed teeth range from 5.3 to 13.4 mm tall with denticle densities of 8–14 per mm. All of the teeth are recurved, fluted, and lack dental caudae (Figs. 1E and 1F). The mesial edge of the teeth changes gradually and follows the mesial-distal long axis. In teeth with preserved crown tips the shape of the denticles remains constant. Of all the taxa included here, NMT RB187 exhibits the greatest degree of recurvature.

### *Pallisteria*

We included 11 in situ teeth from the left and right maxillae and the left premaxilla of a single individual of "*Pallisteria*" (NHMUK PV R36620). These teeth are the largest of all the taxa, ranging from 36.1 to 70.3 mm, and have the lowest density, from two to three per mm. All except the two premaxillary teeth are recurved and all have smooth crowns and lack dental caudae. Most of the teeth have uneven labial-lingual curvature and a mesial edge that changes angles gradually. The mesial carina is offset from the mesiodistal long axis in most the teeth and the denticles remains constant in shape and size along the height of the crown.

### RESULTS

For our linear model we predicted the tooth base shape (ratio of labiolingual base width to mesiodistal base length) using the total apicobasal crown height and the taxonomic affinity of the tooth (base ~ tch + taxon) with the lm() command in base R (Table 2). We found that tooth height was not a significant predictor of base shape (*p* = 0.0933. We used the R package "lsmeans" to further investigate the differences between the species'

**Table 3 Pairwise comparisons of taxa used in the linear model.** (A) Summary table of lsmeans. (B) Summary table of pairwise comparisons. The undescribed pseudosuchian is readily differentiable from most taxa, with the exception of *Parringtonia*. Confidence intervals were generated using a 95% confidence level.

**A**

| Taxon | lsmeans | Standard error | df | Lower CL | Upper CL |
|---|---|---|---|---|---|
| *Asilisaurus* | 1.2354 | 0.1150 | 46 | 1.0039 | 1.4669 |
| Undescribed | 1.6071 | 0.0568 | 46 | 1.4929 | 1.7214 |
| *Nundasuchus* | 1.3348 | 0.0505 | 46 | 1.2332 | 1.4365 |
| "*Pallisteria*" | 1.1105 | 0.1225 | 46 | 0.8640 | 1.3570 |
| *Parringtonia* | 1.4844 | 0.0595 | 46 | 1.3645 | 1.6042 |

**B**

| Contrast | Estimate | SE | df | t ratio | p-value |
|---|---|---|---|---|---|
| *Asilisaurus*—Undescribed | −0.3718 | 0.1148 | 46 | −3.237 | 0.0181 |
| *Asilisaurus*—*Nundasuchus* | −0.0995 | 0.1247 | 46 | −0.798 | 0.9299 |
| *Asilisaurus*—"*Pallisteria*" | 0.1249 | 0.2007 | 46 | 0.622 | 0.9708 |
| *Asilisaurus*—*Parringtonia* | −0.2490 | 0.1127 | 46 | −2.210 | 0.1944 |
| Undescribed – *Nundasuchus* | 0.2723 | 0.0751 | 46 | 3.625 | 0.0062 |
| Undescribed—"*Pallisteria*" | 0.4967 | 0.1571 | 46 | 3.162 | 0.0222 |
| Undescribed—*Parringtonia* | 0.1228 | 0.0677 | 46 | 1.813 | 0.3788 |
| *Nundasuchus*—"*Pallisteria*" | 0.2244 | 0.1343 | 46 | 1.671 | 0.4614 |
| *Nundasuchus*—*Parringtonia* | −0.1495 | 0.0770 | 46 | −1.942 | 0.3107 |
| "*Pallisteria*"—*Parringtonia* | −0.3739 | 0.1631 | 46 | −2.292 | 0.1659 |

tooth shape (Table 3). From this metric NMT RB187 has a significantly higher base shape ratio than all other taxa except *Parringtonia* ($p = 0.3788$).

The sum of variances analysis (Fig. 4) included all known Manda Beds archosauriform taxa with associated dentition and two of the three morphotypes, as only a single tooth of Morphotype C is present in our assemblage. These variances provide a quantification of intraspecific variation in tooth size and shape, and allow for an equal interspecific comparison. Mean variances ranged from a low of 0.02 log units in "*Pallisteria*," two large isolated teeth of *Parringtonia*, and Morphotype B, to a high of 0.145 log units in Morphotype A (Fig. 4).

More useful for visualizing variation than the linear model and lsmeans contrasts are morphospace plots of the teeth from our generically determinate specimens, with the isolated, unidentified teeth added for comparison. There is much overlap in morphospace occupancy, particularly on the left side (shorter height) portion of the graph, although "*Pallisteria*" occupies its own section of morphospace in taller crown heights (Fig. 5). Teeth toward the bottom of the morphospace (lower base ratio) are more rounded and cone-like, whereas those with higher base ratios are more laterally compressed. With size alone two of the Morphotype A teeth fall in "*Pallisteria*" morphospace and the Morphotype C tooth with *Nundasuchus* morphospace contrary to the discrete descriptive predictions. The relationship between base width and mesiodistal base length provides little more

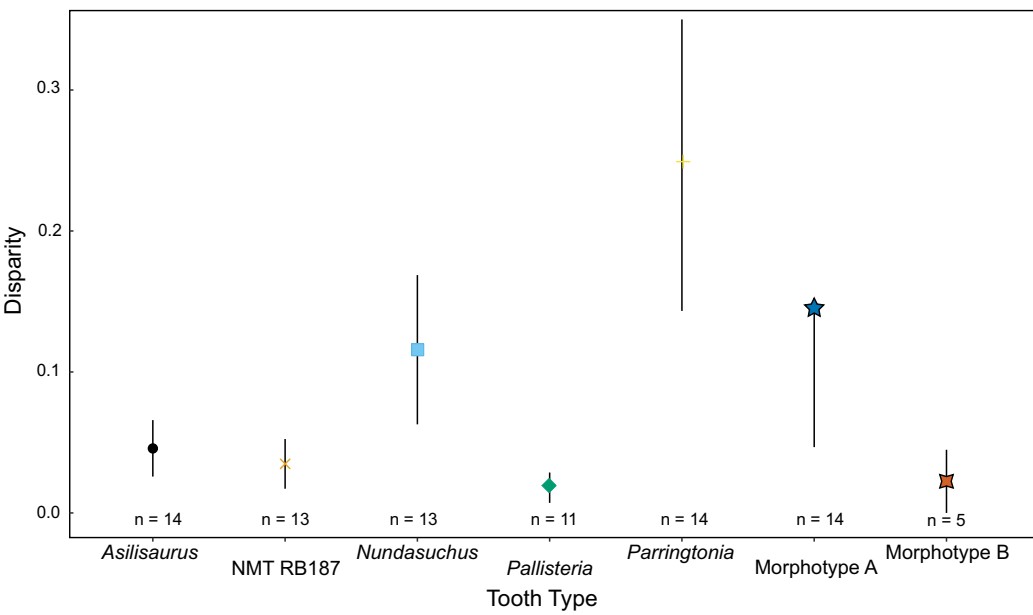

**Figure 4 Disparity of teeth measured by sum of variance.** Disparity divided by taxon or morphotype. The sample sizes reflect the number of teeth with at least one of three measurements that was used to generate the predictive intervals.

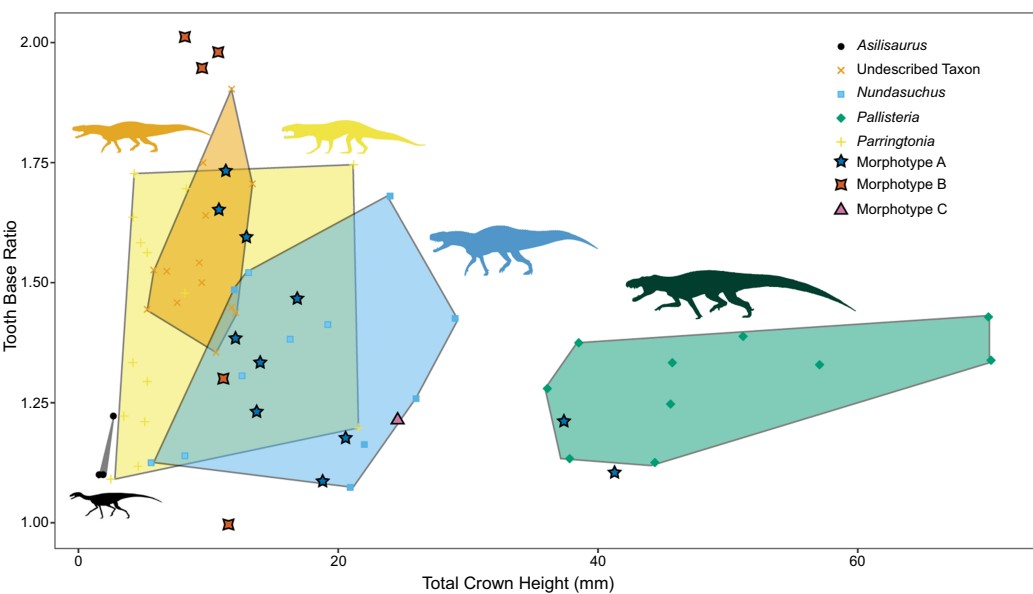

**Figure 5 Relationship between height and base shape of teeth divided by taxon.** The taxonomically unidentified teeth fall within a variety of the morphospaces generated by known taxa, rendering unambiguous referrals impossible. Some genera exhibit much greater variation in base shape ratio than others, potentially indicating a greater level of within-taxon variation.

distinction of the taxa included, and the impact of crown size is still evident (Fig. 6). In general the ratio of base mesiodistal length and labiolingual width follows a linear trend controlled by size. Since size seemed to be the primary driver of differences between

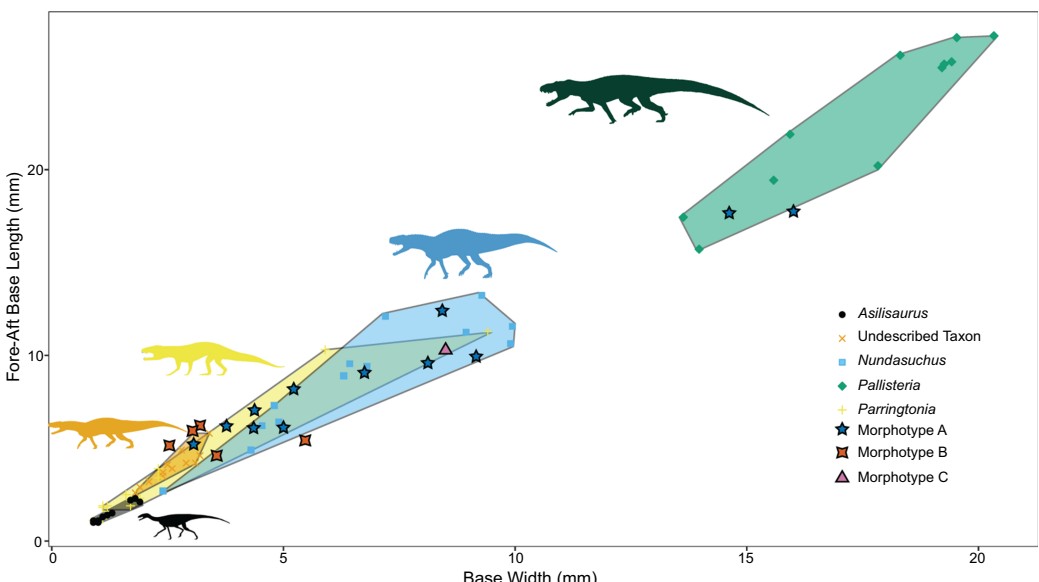

**Figure 6 Relationship between base width and fore-aft base length divided by taxon.** The overall ratio of base shape appears to be highly conserved with little deviation from the general trend. Differentiation between genera appears to be driven primarily by size.

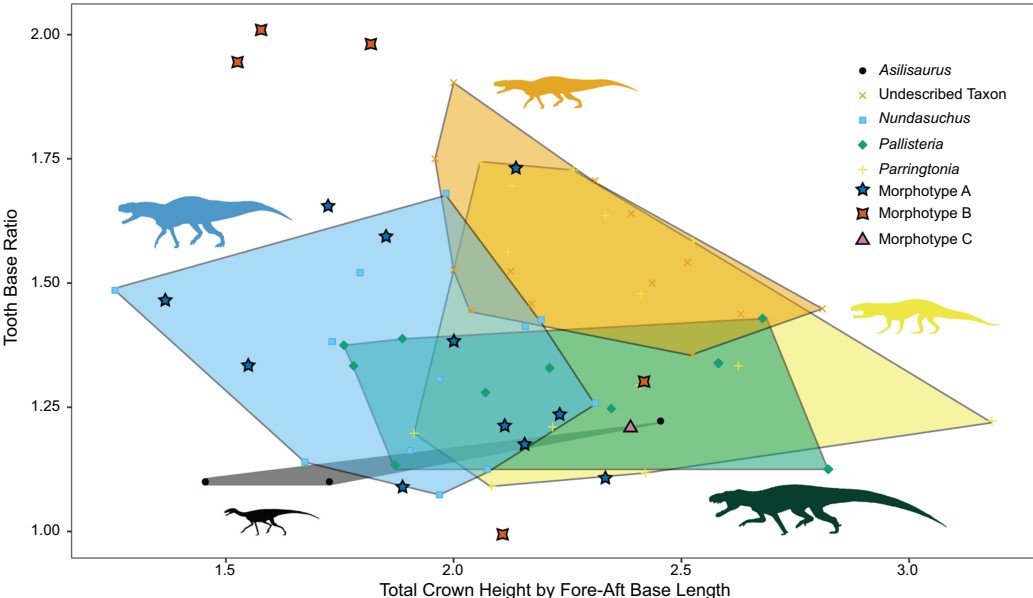

**Figure 7 Relationship between height and base shape of teeth controlled for size.** Morpohotype B teeth appear to have the most labio-lingually compressed teeth. All taxa in our study exhibit a range of tooth base shapes, though some, such as *Parringtonia*, are more variable than others. The morphospaces of all taxa share large amounts of overlap.

taxa, we modified the two previous comparisons in order to control for size. Again we compared the tooth base ratio to total crown height, this time with crown height divided by fore-aft base length to create a ratio for the lateral profile (Fig. 7). This caused even greater overlap in morphospace than the non-size controlled comparison (Fig. 5) as

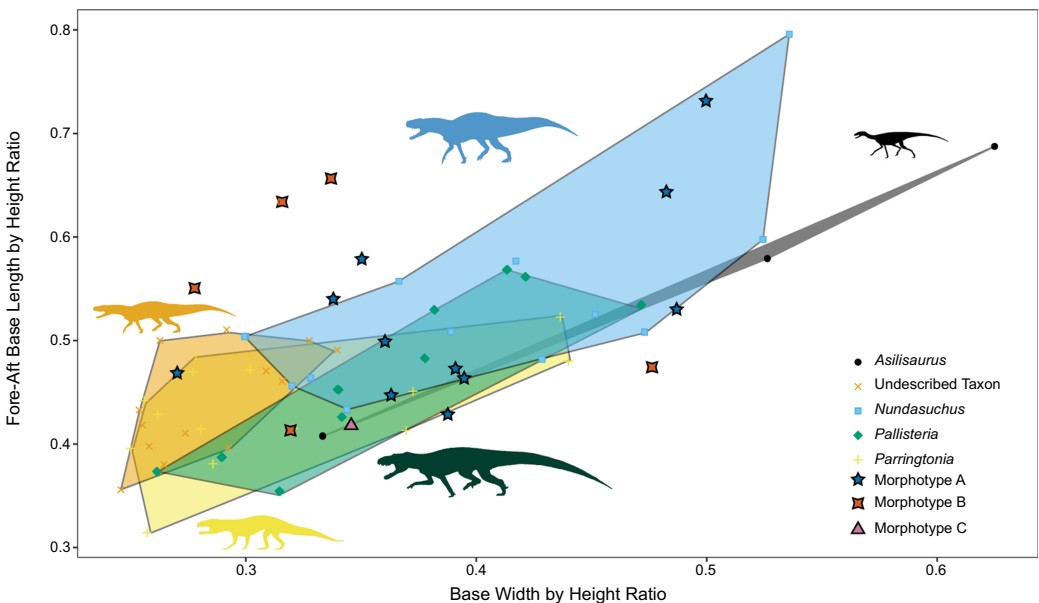

**Figure 8 Relationship between base width and fore-aft base length divided by taxon controlled for size.** There is more deviation from the general trend in base ratio recovered in Fig. 6, but does appear to remain. All taxa have some degree of morphospace overlap with the other taxa.

"*Pallisteria*" was no longer separated from the other taxa (Fig. 7). Similarly, when we compared base width divided by height to fore-aft base length divided by height to control for size the morphospace of taxa increased in overlap (Fig. 8). Again, "*Pallisteria*" was no longer in a distinct region of morphospace (Fig. 8). However, the general trend recovered in Fig. 6 remained even once we controlled for size, though the deviation had increased (Fig. 8).

A total of 21 isolated teeth and 46 in situ teeth of known taxonomic identity were complete enough to be scored for the NMDS analysis. Convex hulls are more differentiated than in the quantitative morphospace, with almost no overlap of *Nundasuchus* with either NMT RB187 or *Parringtonia* (Fig. 9). Overlap of NMT RB187 and *Parringtonia* remains, but most of the isolated teeth fall exclusively within or adjacent to the zone of *Nundasuchus* and "*Pallisteria*" (Fig. 9). The high degree of overlap between *Parringtonia* and NMT RB187 likely reflects their often-shared feature of having parallel ridges (fluting) along the labial and lingual sides of the tooth crown. The only other tooth in the study with fluting is a single example referred to *Nundasuchus*. The use of taxa and morphotype "averages" in traits reveals similar groupings to the complete dataset, with average morphotype scores between those of known taxa (Fig. 10).

## DISCUSSION

We present the first quantitative description of a Middle Triassic archosauriform tooth assemblage, which reveals substantial conservation of tooth morphology at the beginning of the archosaur radiation. Intraspecific variation appears to be as great, if not greater, than interspecific variation. Morphotype A displays the greatest variance in tooth size in the

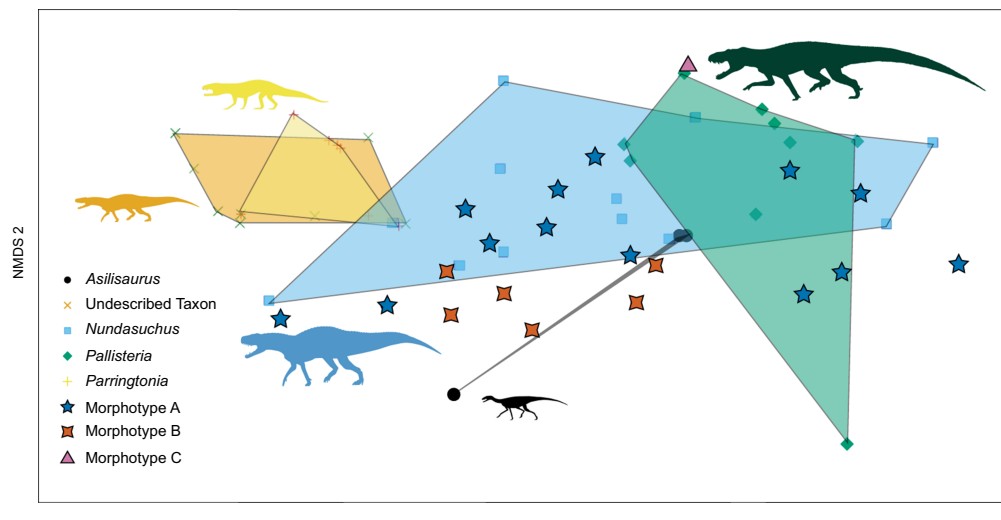

**Figure 9 Ordination plot of first two major NMDS axes of tooth morphospace.** Colored, transparent polygons represent the convex hulls of known taxa. Each point represents a separate tooth scoring. *Parringtonia* and NMT RB187 (undescribed taxon) share almost the same morphospace and there is substantial overlap between *Nundasuchus* and "*Pallisteria*" also. Morphotype A appears to be more variable than Morphotype B, which is clustered closer together within a subsection of overall Morphotype A morphospace. The proximity of *Asilisaurus* to *Nundasuchus* and "*Pallisteria*" is likely an artifact of incomplete scorings for *Asilisaurus* teeth.

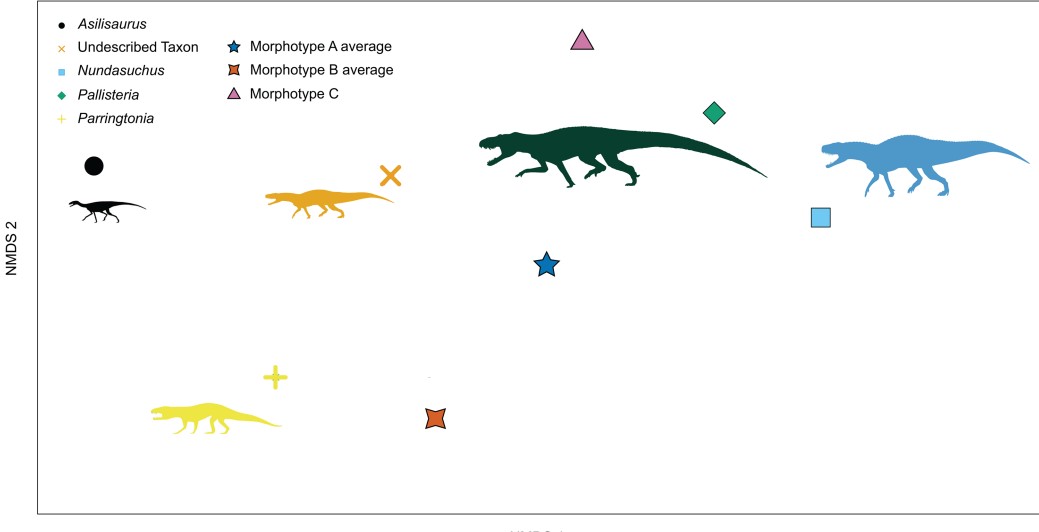

**Figure 10 Ordination plot of first two primary NMDS axes of tooth "averages" morphospace.** Taxa scoring represent "average" scores for each taxon. Only Morphotype C is represented by a single tooth. When using typical scores for taxa we find *Asilisaurus* is no longer near *Nundasuchus* and "*Pallisteria*" morphospace, but on the far side of ordination space.

sample, although *Nundasuchus* has a very similar sum of variance structure (Fig. 4). Driving at least part of the pattern we see in our disparity analysis is whether more than a single individual of a given taxon is included in our study. For example, NMT RB187, "*Pallisteria*," and *Parringtonia* all display low disparity, but our sample includes only

elements from a single individual of each taxon, whereas the *Nundasuchus* sample includes in situ teeth from one lower jaw (the holotype specimen) and associated isolated teeth assigned to the holotype (Fig. 4). Although two of the isolated teeth from our assemblage fall exclusively within the "*Pallisteria*" quantitative morphospace, most of the isolated teeth fall within a zone of overlap between *Nundasuchus*, NMT RB187, and *Parringtonia* (Fig. 5). Much of this quantitative variation reflects body size (Fig. 6). *Nundasuchus* and "*Pallisteria*" are much larger than the other taxa, which helps to differentiate their morphospace from that of smaller-bodied taxa. *Asilisaurus* is the smallest taxon in our sample, but there is postcranial evidence of a larger silesaurid in the Lifua assemblage (possibly a very large individual of *Asilisaurus*; *Barrett, Nesbitt & Peecook, 2015*) that would be comparable in size to *Nundasuchus* and "*Pallisteria*." Recovery of teeth from silesaurid individuals of this larger size might reduce some of the differentiation between them, *Nundasuchus*, and "*Pallisteria*" though we would still expect silesaurid teeth to be smaller relative to the same body size. By controlling for size in our analysis on linear measurements, much of the morphospace differentiation, particularly for the large-bodied "*Pallisteria*" (Figs. 7 and 8), was lost. Although size can be an important consideration when reconstructing diet, this demonstrates that when attempting to reconstruct general tooth shape distinctions between taxa size should be controlled. Additionally, it is worth noting that our study lacks ontogenetic data. Because we are unable to tell the age of the isolated teeth and the maximum body size, and the in situ specimens do not capture an ontogenetic series, we may miss important disparity changes from ontogeny. The inclusion of isolated teeth should capture some of the ontogenetic variation not found in our in situ specimens, yet likely not the entire range.

The NMDS ordination improves the differentiation of taxa, with *Asilisaurus* and the large-bodied predator "*Pallisteria*" more clearly separated from the still overlapping undescribed pseudosuchian, and *Parringtonia* and *Nundasuchus* exhibiting wide variation in morphospace overall, bridging the space between all taxa, and overlapping a substantial part of "*Pallisteria*" morphospace (Fig. 9). These results identify two general areas of morphospace, one shared by the undescribed pseudosuchian and *Parringtonia* and the other by *Nundasuchus* and "*Pallisteria*." The teeth of *Parringtonia* and the undescribed pseudosuchian share several features, notably presence of fluting, a mesial carina along the midline tooth axis, and a high denticle density (≥3 per mm). By contrast, *Nundasuchus* and "*Pallisteria*" teeth lack fluting, possess an offset mesial carina, unequal labial/lingual curvature, and have a low denticle density (<3 per mm). This result is further supported when the average or typical score of each taxon is used, with NMT RB187, *Asilisaurus*, and *Parringtonia* clustering together versus *Nundasuchus* and "*Pallisteria*" on the other side of morphospace (Fig. 10). Given that many of the isolated teeth resemble those of *Nundasuchus*, it is not surprising that most of the isolated teeth fall within the convex hull defined by *Nundasuchus* (Fig. 9). We cannot, however, definitely assign these teeth to *Nundasuchus* due to the overlap in discrete characters among our included taxa.

Our results using both methods demonstrate that many of the isolated teeth resemble those from currently recognized taxa. However, several teeth fall outside of the

morphospace defined by known taxa and could indicate either intraspecific variation (due to heterodonty or ontogeny) or could represent other, as yet unsampled, taxa. Our methodologies are flexible and the datasets can incorporate additional specimens as they are excavated, so these approaches could be applied to other tooth assemblages throughout the Triassic across a broad range of spatial, temporal, and taxonomic scales.

### Ecological differentiation

There are some hints of dietary separation between large- and small-bodied archosaurs based on minor changes in tooth morphology and consideration of body size. However, our results, which show high degrees of overlap in tooth morphology suggest that ecological differentiation, at least in diet, appears to lag behind lineage diversification, at least with respect to Manda archosauriforms. Four of the five recognized taxa included here possess ziphodont dentitions (= labiolingual narrow crown (labiolingual width <60% of mesiodistal length), recurved, typically serrated carinae, and no constriction at the cervix sensu, *Hendrickx, Mateus & Araújo, 2015*) indicative of a carnivorous diet. Only *Asilisaurus* differs in possessing a conidont dentiton (= conical crowns with small denticles or no denticles, and typically fluted sensu, *Hendrickx, Mateus & Araújo, 2015*). Conidonty is present in spinosaurids, many crocodylians, marine reptiles, and pterosaurs (*Hendrickx, Mateus & Araújo, 2015*) and has been linked to piscivory. Following this criterion *Asilisaurus* would be categorized as a potential piscivore. However, dietary reconstructions of *Silesaurus opolensis*, another silesaurid possessing similar dentition to *Asilisaurus*, have been herbivorous or omnivorous based upon dental microwear (*Kubo & Kubo, 2014*) or insectivorous based upon coprolites (*Qvarnström et al., 2019*). Thus, in the Manda Beds tooth assemblage, there are two large-bodied carnivores (*Nundasuchus* and "*Pallisteria*"), two small-bodied carnivores (*Parringtonia* and an undescribed pseudosuchian), and one small-bodied, non-carnivore (*Asilisaurus*). The Middle Triassic Manda Beds may, therefore, be capturing the beginning of the "Explosive Phase" of *Simpson's (1944)* theoretical model as lineages split and begin to move toward new adaptive zones. Further tooth assemblages will need to be evaluated to see if this is a broader trend that holds across the Triassic archosaur radiation. We posit that the qualitative NMDS ordination method gives us the necessary lens for testing this hypothesis.

## CONCLUSIONS

Simple quantitative measures of tooth shape were of limited use in characterizing the Middle Triassic Manda Beds archosauriform tooth assemblage because of the highly conserved morphology of many specimens. Instead, an ordination based on discrete characters provided a more effective means of differentiating the teeth of distinct taxa. Nevertheless, we found little evidence for significant ecological differentiation of tooth shape between the five taxa included in our study. Most isolated teeth ($N = 17/21$) fall within the spectrum of recognized taxon variation, and the remainder represent either unsampled taxa or unsampled intraspecific variation. We interpret this as evidence that ecological disparity in diet lagged behind lineage diversification during the archosauriform radiation following the PTME. This in turn indicates that at least in some adaptive

radiations, the process of ecological breadth expansion in a clade may be separate from the earlier lineage diversification as proposed by *Simpson (1944, 1953)*.

Our relatively simple metrics can be used to describe subtle differences in tooth morphology. These objective methods for grouping teeth provide a complimentary method for assigning teeth to dietary roles, a practice that typically relies on qualitative comparisons to the teeth of extant taxa of known diet (*Fraser & Walkden, 1983*; *Sander, 1999*; *Barrett, 2000*; *Hungerbühler, 2000*) or other fossil taxa (*Dzik, 2003*; *Hendrickx, Mateus & Araújo, 2015*; *De Oliveira & Pinheiro, 2017*). Furthermore, the methods applied herein provide an evaluation of ecological disparity that is separate from the features used in phylogenetic analyses, so that we can compare these two evolutionary phenomena independently. This method is readily transferable to tooth assemblages from other localities pertaining to any vertebrate clade. Our next step will be to apply this technique to richer Middle Triassic sites, as well as Late Triassic sites, to understand how morphological and ecological diversity changed during the early stages of the archosauriform radiation.

## INSTITUTIONAL ABBREVIATIONS

**NHMUK**    Natural History Museum, London, UK
**NMT**      National Museum of Tanzania, Dar es Salaam, Tanzania.

## ACKNOWLEDGEMENTS

We thank Dr. Kate Langwig and Dr. Josef Uyeda for their assistance and advice with R Studio, and the Virginia Tech Paleobiology Research Group for helpful comments.

### Funding

Manda Beds specimens were collected with funding provided by the National Science Foundation EAR 1337291, National Geographic Society grants 7787-05 and 8962-11 grants, with additional support from NSF DBI-0306158 and The Grainger Foundation. Devin K. Hoffman is supported by the NSF GRFP. The funders had no role in study design, data collection and analysis, decision to publish, or preparation of the manuscript.

### Grant Disclosures

The following grant information was disclosed by the authors:
National Science Foundation: EAR 1337291.
National Geographic Society: 7787-05 and 8962-11.
NSF DBI-0306158.
The Grainger Foundation.
NSF GRFP.

### Competing Interests

The authors declare that they have no competing interests.

## Author Contributions

- Devin K. Hoffman conceived and designed the experiments, performed the experiments, analyzed the data, prepared figures and/or tables, authored or reviewed drafts of the paper, approved the final draft.
- Hunter R. Edwards conceived and designed the experiments, performed the experiments, prepared figures and/or tables, authored or reviewed drafts of the paper, approved the final draft.
- Paul M. Barrett performed the experiments, contributed reagents/materials/analysis tools, authored or reviewed drafts of the paper, approved the final draft.
- Sterling J. Nesbitt conceived and designed the experiments, performed the experiments, contributed reagents/materials/analysis tools, authored or reviewed drafts of the paper, approved the final draft.

## Field Study Permissions

The following information was supplied relating to field study approvals (i.e., approving body and any reference numbers):

Tanzania's Ministry of Natural Resources and Tourism and the Tanzania Commission for Science and Technology (COSTECH).

## Data Availability

The quantitative measurements of the teeth and the scoring of discrete characters are available as Supplemental Files.

All specimens except 'Pallisteria' (NHMUK PV R36620) are currently housed at Virginia Tech Department of Geosciences and will be permanently reposited in the National Museum of Tanzania. NHMUK PV R36620 is permanently housed in the Natural History Museum (UK). Accession numbers are: NMT RB807–NMT RB821, NMT RB825, NMT RB826, NMT RB828, NMT RB831, NMT RB192, NMT RB150, NMT RB151, NMT RB48, NMT RB426, NMT RB837, NMT RB187, NMT RB159 and NHMUK PV R36620.

## Supplemental Information

Supplemental information for this article can be found online at http://dx.doi.org/10.7717/peerj.7970#supplemental-information.

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
