# Peer review of "Reconstructing the archosaur radiation using a Middle Triassic archosauriform tooth assemblage from Tanzania"

_PeerJ, doi:10.7717/peerj.7970_

## Round 0.1 · original submission · Major Revisions

Dear Dr. Hoffman and colleagues:

Thanks for submitting your manuscript to PeerJ. I have now received two independent reviews of your work, and as you will see, the reviewers raised some concerns about the research. Despite this, these reviewers are optimistic about your work and the potential impact it will have on archosaur evolution and systematics. Thus, I encourage you to revise your manuscript, accordingly, taking into account all of the concerns raised by both reviewers.

Please be sure to access the marked-up manuscripts provided by reviewers 1 and 2, who both took the time to provide edits.

While the concerns of the reviewers are relatively minor, this is a major revision to ensure that the original reviewers have a chance to evaluate your responses to their concerns.

I look forward to seeing your revision, and thanks again for submitting your work to PeerJ.

Good luck with your revision,

-joe

·

Basic reporting

I have reviewed the manuscript “Reconstructing the archosaur radiation using a Middle Triassic archosauriform tooth assemblage from Tanzania” by Devin Hoffman and collogues. This is an important contribution to the field of paleontology and outlines a significant method to quantify dental form as well as thorough descriptions of a diversity of Triassic dentitions from the Manda Beds in Tanzania. This research is appropriate for PeerJ and is an exciting contribution to the field of paleontology. This work also has exciting implications for paleoecological reconstruction in extinct reptiles and will hopefully open a novel, non-destructive avenue for dietary reconstruction.
Overall, the manuscript is well written, with only minor grammatical errors. It begins with an informative introduction, which clearly provides necessary background information and outlines the primary goals of the project. Aside from a few grammatical changes outlined in attached PDF, I would suggest adding a sentence on previous work that’s used quantitative methods to address tooth shape including work by D'Amore et al. (2019), Larson et al. (2016) (who is cited later), and my own 2019 work. Around line 82 may be a potential starting point.
As for other general requirements, the manuscript is largely in line with PeerJ standards. The manuscript is structured correctly and clearly. Unless I am mistaken, the references need to be reformatted to meet PeerJ guidelines. In fact, the section is misidentified as ‘Works Cited’ as opposed to ‘References’ as in other PeerJ publications. Additionally, some references in the Works Cited are not in the manuscript (e.g., Larson and Scanlon’s work, for instance). All figures are clear and necessary, adding important information to the manuscript. Minor edits are attached on the PDF. Finally, the authors make available the raw data used in the study, which is greatly appreciated.

Experimental design

My major concern with this paper is found in the experimental design. The first is that they may be biasing the total disparity investigated by only looking at teeth that look like those that belong to known archosauriforms. This is somewhat unavoidable for a study like this and does not negate the importance of the research, but it should be mentioned at some point in the paper.
The second concern is with how size is dealt with in the analyses. Figures 5 and 6 clearly show the primary axis of discrimination is size. The disparity measurements are log-transformed, but what about the linearly analyses? The reason I think this is an important distinction is that size is a part of what shape isn’t, which is slightly confusing, but by increasing size of something you’re not changing shape. So, when size is the primary driver in morphology studies you’re not really measuring shape, just size. To address this, you should normalize the data by size, including by adding additional measurement and dividing them by tooth height or something to that effect. Similarly, in the disparity analysis, Parringtonia is broken up into large and small groups, but if the data are log-transformed it should not need to be. Other taxa are not broken up and I think that if you’re going to present the data separated then it should also be figured as a complete dataset as well. The NMDS analysis addresses this by looking at only discrete traits and I think it would be beneficial to compare this result with a size-normalized linear result.
Aside from these worries, the experimental design is excellent. The research questions and goals are well defined. The methods are clear and thorough, and, to my knowledge, the analyses are performed to a high technical and ethical standard.

Validity of the findings

All findings are presented clearly and concisely, and this work will be an important contribution once revised. All data and results are presented, and when statistically significant, highlighted. Speculation on both ecology and evolutionary theory are done carefully and well within reason. Again, my concerns are with much of the analyses summarizing tooth size and not shape, but these worries can be taken care of if size is removed from the linear data. Additionally, the conclusions should mention adaptive radiation, as much of the introduction is based on exploring this topic using Triassic archosauriforms. Otherwise, this is a great paper!

·

Basic reporting

In general, basic reporting is sound. I have made one or two suggestions of adding additional references, but the article is well-cited overall. The structure of the article makes sense, and it is clearly self-contained. I would suggest increasing the number of citations to your figures substantially, particularly when describing the tooth morphology of each taxon and unknown morphotype. Additionally, please cite specimen numbers, especially when referring to individual teeth that deviate from the typical morphology.

Experimental design

The experimental methods are well-described, and the research question is very relevant and meaningful for paleontology. It is a difficult but important task to identify proxies for the taxonomy of isolated and fragmentary fossil specimens in the absence of unambiguous apomorphies, so projects like yours have the potential to add a lot to our understanding diversity in deep time. I also suggest that you make it clearer in your introduction how this archosauriform tooth assemblage relates to the early radiation of archosaurs.

Validity of the findings

In general, results and conclusions are well written and relevant to the theme of the paper; however, I would like to see you address the relative body sizes of your known taxa. I.e., how do we know that the discrete morphospaces you identify and depict in Figures 4-8 aren't simply related to body size? Additionally, it isn't clear to me why _Parringtonia_ is split in Figure 4 but not in subsequent figures.

Additional comments

Dear Devin, Hunter, Paul, and Sterling:

This is an important contribution, and as someone who also works on lots of fragmentary and isolated specimens, I am always excited to see new methods for tracking potential hidden taxonomic diversity in the fossil record.

A few other minor issues to correct that don't really fit in the above categories:
In Figures 4, 5, 6, and 7 "Morphotype B" is mis-spelled "Morphtype B."
All of your references are not formatted correctly for PeerJ (see: https://peerj.com/about/author-instructions/#reference-format).

In addition to the points above, I have made minor editorial comments and stylistic suggestions to your manuscript, so please see the PDF attached to my review.

I'm suggesting your article is accepted once you address the minor revisions I've suggested.

Good work, and thank you for the opportunity to review your paper!

Best,
Benn

---

## Round 0.2 · accepted · Accept

Dear Dr. Hoffman and colleagues:

Thanks for re-submitting your revised manuscript to PeerJ, and for addressing the concerns raised by the reviewers. I now believe that your manuscript is suitable for publication. Congratulations! I look forward to seeing this work in print, and I anticipate it being an important resource for research on archosaur evolution and systematics.

Thanks again for choosing PeerJ to publish such important work.

-joe

·

Basic reporting

No comment

Experimental design

No comment

Validity of the findings

No comment

Additional comments

The authors have made all suggested changes and the paper is greatly improved. As it stands, I think the manuscript is ready for publication!

·

Basic reporting

No comment, see previous review. All concerns were addressed by the authors in their revisions.

Experimental design

No comment, see previous review. All concerns were addressed by the authors in their revisions.

Validity of the findings

No comment, see previous review. All concerns were addressed by the authors in their revisions.

Additional comments

Dear Devin, Hunter, Paul, and Sterling,

I think your revisions look good and am recommending your article for publication.

Good work, and thanks again for the opportunity to review this!

-Benn